# ^13^C-NMR Chemical Shifts in 1,3-Benzazoles as a Tautomeric Ratio Criterion

**DOI:** 10.3390/molecules27196268

**Published:** 2022-09-23

**Authors:** Efrén V. García-Báez, Itzia I. Padilla-Martínez, Alejandro Cruz, Martha C. Rosales-Hernández

**Affiliations:** 1Laboratorio de Química Supramolecular y Nanociencias, Instituto Politécnico Nacional-UPIBI, Av. Acueducto s/n, Barrio la Laguna Ticomán, Mexico City 07340, Mexico; 2Laboratorio de Biofísica y Biocatálisis, Sección de Estudios de Posgrado e Investigación, Escuela Superior de Medicina, Instituto Politécnico Nacional, Plan de San Luis y Salvador Díaz Mirón s/n, Casco de Santo Tomás, Mexico City 11340, Mexico

**Keywords:** benzimidazoles, ^13^C NMR, tautomeric equilibrium, mesomery, pyridine nitrogen atom, pyrrole nitrogen atom

## Abstract

Benzimidazole is an important heterocyclic fragment, present in many biologically active compounds with a great variety of therapeutic purposes. Most of the benzimidazole activities are explained through the existence of 1,3-tautomeric equilibrium. As the binding affinity of each tautomer to a protein target depends on an established bioactive conformation, the effect of tautomers on the ligand protein binding mechanism is determinant. In this work, we searched and analyzed a series of reported ^13^C-NMR spectra of benzazoles and benzazolidine-2-thiones with the purpose of estimating their tautomeric equilibrium. Herein, several approaches to determine this problem are presented, which makes it a good initial introduction to the non-expert reader. This chemical shift difference and C4/C7 signals of benzimidazolidine-2-thione and 1-methyl-2-thiomethylbenzimidazole as references were used in this work to quantitatively calculate, in solution, the pyrrole–pyridine tautomeric ratio in equilibrium. The analysis will help researchers to correctly assign the chemical shifts of benzimidazoles and to calculate their intracyclic or exocyclic tautomeric ratio as well as mesomeric proportion in benzimidazoles.

## 1. Introduction

The tautomerism phenomenon is considered as a dynamic equilibrium between interconvertible structural isomers, named tautomers, with the migration of one atom or group. When a hydrogen atom migrates, the phenomenon is known as prototropy; however, other groups such as alkyl, aryl, acyl, cyano, halogens, amines, and nitro, as well as methals can migrate; Figure 1 [1].

Nuclear magnetic resonance (NMR) spectroscopy is one of the most useful techniques to study tautomerism. The signals observed in the spectra depend on the activation energy between the tautomers, which determines their lifetimes and chemical shift difference (Δδ). When lifetimes are long, compared with 1/Δδ, a slow exchange regime gives rise to separate narrow signals for each of the tautomers. In this case, integration of the ^1^H-NMR signal intensities is the method of choice to study the tautomerism. For shorter lifetimes, the exchange is raised, leading to line broadening. Then, high magnetic fields and lower temperatures can be used to achieve slow exchange conditions. Interpolation is the method of choice when lifetimes are shorter than 1/Δδ, and fast exchange and signal coalescence predominate to give averaged narrow signals, but the NMR chemical shifts of the individual tautomers remain unknown. Four techniques have been used to solve this problem: (1) the use of blocked derivatives of individual tautomer, replacing the tautomeric proton by a methyl group, and performing a correction for the substituent effect; (2) the use of model compounds that exclusively exist in one tautomeric form; (3) the use of the properties measured in the solid state where only one tautomer exists, but phase effects should be considered; and (4) the use of theoretically calculated properties as GIAO absolute shielding, however, solvent effects are difficult to estimate.

Benzazoles (BZs) are constituted by a benzene ring fused to an oxazole (BO), thiazole (BT), or imidazole (BI) ring; left in Figure 1. A typical kind of tautomerism in BI is the relocation of a proton, known as annular tautomerism (a). BZs containing an exocyclic heteroatom, where a proton can migrate from cyclic nitrogen to an exocyclic heteroatom, retaining the aromaticity, represent an exocylic tautomerism (b); right in Figure 1.

In commercial databases, the presence of tautomeric duplicates has been found, and both tautomers are offered as different products [2]. On the other hand, different names for tautomers based on the IUPAC rules have been used [3]. In general, compounds with well-known tautomerism are named as the predominant tautomer at equilibrium. For example, 2-mercaptobenzimidazole (MBI) is named as benzimidazolidinethione (BIT) or 1,3-dihydro-2*H*-benzo[d]imidazole-2-thione (X = NH, Y = S); Figure 1.

The tautomeric interconversion with a low-energy barrier, in general, displays averaged signals in the NMR spectrum. To slow down the interconversion rate in solution, the following strategies are used: lowering temperature and/or the use of hexamethylphosphoramide-d_18_, which prevents the formation of hydrogen bonds. Thermodynamic and kinetic aspects of the tautomeric equilibrium have been explored using dynamic NMR studies [4,5,6] Because of the wide range of chemical shifts and high sensitivity, ^15^N-NMR spectroscopy has been used to study the tautomerism of nitrogen in heteroaromatic compounds [7]. Solid-state NMR experiments [4] 4, X-ray crystallography [8,9,10], and ^15^N-NMR studies in DMSOd_6_ for both BIT and 1-MeBIT revealed that the thione form is the predominant tautomer [11]. Recently, Pandey et al. theoretically studied 5-MeOBIT using B3LYP methods with a 6-311++G (d, p) basis set. A comparison between the experimental and calculated structure as well as the calculated and experimental ^1^H and ^13^C chemical shifts showed a good correlation with the thione isomer [12].

Among the BZs, benzimidazole (BI) is the most known and the one whose tautomeric equilibria have been widely studied [4,5,6,7]. It has been stablished that, when a large substituent is bonded at the N1 position in 2-heterosubstituted BIs, two isomeric compounds exist because of the annular tautomerism. BI is an important heterocycle, present in several biologically active compounds. It has been found in progesterone receptor antagonists [13], luteinizing hormone-releasing hormone antagonists (leuprolide, goserelin, triptorelin) [14,15], antiviral (enviradine) [16,17], antiprotozoal [18,19], antimicrobial [20,21,22], analgesic and anti-inflammatory [23], anticonvulsant, antidiabetics [24], anthelmintics (albendazole, mebendazole, and thiabendazole), proton pump inhibitors (omeprazole, lansoprazole, and pantoprazole), antihistaminic (astemizole), and antihypertensives (candesartan, cilexitil, and telmisartan), among others.

The binding affinity of each tautomer to a protein target depends on an established bioactive conformation. Therefore, the method proposed herein to calculate in solution the prototropic ratio in BZs, on the basis of the electronic effect of pyrrole like atom (NH = N_pr_) and pyridine like atom (N_pd_) on C4 and C7 chemical shift resonances, is very valuable. On the other hand, the method is also helpful to correctly assign the carbon atoms on the benzene ring.

## 2. Electronic Effects of Heteroatoms on Aromatic Rings

### 2.1. Influence of Benzene Ring Substituents in the ^13^C NMR Chemical Shifts

For clarity reasons, a brief introduction about the inductive and electronic effects of heteroatoms (X = N, O, S) on the ^13^C chemical shifts (δ) of the benzene ring [25] is required. Then, it is interesting to note that both NH_2_ and OH groups deshield the carbon atom to which they are attached, C*_i_* (*ipso*), in 18.2 and 26.9 ppm, respectively, owing to a strong inductive effect. Conversely, the electronic delocalization of the free electron pairs of N and O atoms produces a considerable shielding electronic effect on C*_o_* (*ortho*) in 13.4 and 12.8 ppm, respectively, and C*_p_* (*para*) in 10.0 and 7.4 ppm, respectively. The SH group shifts C*_i_*, C*_o_*, and C*_m_* (*meta*) to high frequencies in only 2.1 ppm, 0.7 ppm, and 0.3 ppm, respectively, but C*_p_* to low frequency by 3.2 ppm. All comparisons are referenced to the δ of the benzene ring (128.0 ppm).

In this sense, the δC4 in the dibenzofused heteroaromatic compounds **1**–**3** (Table 1) appears at 111.6, 110.8, and 122.9 ppm for X = O, NH, and S, respectively [25]. As expected, the inductive effect of the heteroatom on C*i* shifts δ to high frequencies, in agreement with the electronegativity of the heteroatom. 

### 2.2. Effect of Heteroatoms on ^13^C-NMR of Benzazoles

From the reported ^13^C NMR data, listed in Table 2, it can be seen that heteroatom (X) exerts an inductive effect, deshielding C2, C3a, and C7a; Figure 2. On the contrary, the lone electron pair (lep) of the heteroatom X exerts a protective effect, shielding C7 through the electronic delocalization of the lep into the benzofused ring. In the case of BO (**1**), the oxygen atom shifts C7 to 110.8 ppm, while C4 at 120.5 ppm indicates that N3 (pyridine like nitrogen) exerts a very small electronic effect on this carbon because the lep are located out of the conjugated π-electron system. In BT 2, the δ of C7 at 123.6 and δC4 at 121.8 ppm are indicative of the very small shielding effect known for sulfur and pyridine like nitrogen atoms. For BI **3**, X = NH, a set of four signals appear in the ^13^C NMR spectra, independently from the solvent used. The observed *C2* symmetry is the result of tautomeric equilibrium, making C5 magnetically equivalent to C6, C4 to C7, and C3a to C7a. In BIs, the tautomeric species are the result of intermolecular proton transfer in solution from N1 to N3. Then, the signals of C4 and C7 appear at approximately 115.0 ppm, an intermediate δ value between a pyridine like nitrogen (N3) and pyrrole like nitrogen (N1). This observation can be interpreted as both nitrogen atoms in BIs being present in approximately a 1:1 proportion.

In general, in BIs 3, the tautomeric equilibrium is slowed down at low temperatures and/or the increasing polarity of solvents. In polar solvents, the equilibrium exchanging rate decreases and the four signals of BIs are broadened or sometimes disappear from the spectra [4]. The use of the high polar solvent HMPA-d_8_ makes the equilibrium exchange slow enough to observe seven signals of the most stable tautomer, in the ^13^C-NMR spectrum. For instance, the spectrum of BI, recorded in HMPA-d_8_, shows two different broad signals for C4 and C7 in 118.7 and 110.9 ppm, respectively, in close correspondence with the observed values in the solid state (CPMAS), at 119.9 and 112.2 ppm, respectively [4]. 

Within the above exposed background and the analysis of reported ^13^C-NMR data, herein, we expose the usefulness of C4 and C7 chemical shifts as a criterion to estimate the position of the tautomeric equilibrium in BIs.

## 3. Estimation of the Anular Tautomerism on Benzimidazol

The chemical shift reference values for the estimation of the tautomeric equilibrium were set using the 1-methyl-benzimidazole (1-MeBI, **4**) as a model compound. In case **4**, the tautomeric equilibrium is non-existent; therefore, a set of seven narrow signals are observed in CDCl_3_. The δC4 appears at 120.4 ppm, characteristic of an N_pd_ effect, whereas the δC7 at 109.5 ppm is characteristic of an N_pr_ effect. In DMSOd_6_, the δC4 and δC7 values are 119.2 and 110.1, respectively. Then, a good approximation for 100% of the pyridine like character of δC4 and 100% of the pyrrole like character of δC7 is 120.0 and 110.0 ppm, respectively. Furthermore, the tautomeric proportion in BIs can be calculated on a base equal to 120.0 ppm for δC4_ref_ (N_pd_) and 110.0 ppm for δC7_ref_ (N_pr_) as reference values; Figure 2.

The tautomerism in BIs can be calculated considering that the observed chemical shift δ_obsC4/C7_ is the result of the averaged C4 and C7 chemical shifts weighted by the respective molar fraction contributions x_pd_ (pyridine character) and x_pr_ (pyrrolic character). Equations (1)–(3) can be used to calculate the molar fractions of each tautomer; Figure 3.
δ_obs_ = (x_pd_δC4ref) + (x_pr_δC7ref)(1)
x_pd_= (δ_obs_ − δC7ref)/(δC4_ref_ − δC7_ref_)(2)
x_pr_ = 1 − x_pd_(3)

From the reported ^13^C data for BI **3**, as well as their protonated and deprotonated derivatives, in DMSOd_6_ as solvent, Table 3 [30], the tautomeric proportion can be calculated. For BI, the δ_obs_ value is 116.3, then the estimation is 63% of the pyridine character (+N3H). In the case of δ_obs_ for protonated BI (118.3 ppm), a small effect of positive charge on nitrogen atoms shifts this signal to high frequencies in approximately 2.0 ppm. On the contrary, any effect of the negative charge on nitrogen in deprotonated BI was observed. 

This approximation can be applied to 2-aminomethylbenzimidazole (2-AMBI); Figure 2. In the work of Sierra-Centeno et al. [31], the acid–base equilibrium constants of 2AMBI were determined in aqueous solutions at 25 °C through ^13^C NMR spectroscopy, potentiometric and spectrophotometric techniques, as well as through theoretical methods. ^13^C NMR spectra were recorded for solutions of 0.1 M of 2-AMBI at pH values from 1.1 to 13.2, containing 10 % *v/v* of D_2_O. During the titration process, only five signals were observed. The sets of C5 and C6 signals were shifted approximately 7.0 ppm to lower frequencies and C2 and C3a/C7a in 10.0 and 15.5 ppm, respectively, to higher frequencies, whereas the set of C4/C7 signals remained almost constant at 114.5 ppm. Therefore, the BI heterocycle retains its 45% pyridine character independently from the pH.

## 4. Tautomers and Mesomers in 1- and 2-Substituted Benzimidazoles

### 4.1. Effects of the Contribution of Mesomeric Structures in 1-Substituted Benzimidazoles

This approximation can also be used to estimate the participation of mesomeric forms to the structure. Approximately the same pyrrole effect on C7 (109.5 ppm) for **1-**MeBI **4** compared with C7 (110.4 ppm) of 1-phBI **5** in CDCl_3_ was observed; Table 4. However, in the spectrum of **5** obtained in DMSOd_6_, C7 appeared in 116.5 ppm, which can be explained as a solvent effect that stabilizes the electronic delocalization of the lep of N1 to the N-phenyl ring through the mesomeric form **5b**, deshielding C7. Considering the δC7 value of 110 ppm to the resonance form **5a**, the contribution of the resonance form **5b** is estimated as 35% in DMSOd_6_. This effect is also observed for acylBI **6** even in CDCl_3_, whose mesomeric structures can be represented as **6a** and **6b** (approximately 50%).

### 4.2. Effects of the Contribution of Tautomers in 2-Substituted Benzimidazoles

In general, the δC2 in ^13^C NMR spectra for 2-alkyl-benzimidazoles (2-AkBIs) **7** is shifted to higher frequencies in 10–15 ppm compared with C2 in BI; Table 5. Moreover, C3a/C7a and C4/C7 appeared as broad signals with a tendency to coalesce with an increasing alkyl chain length [35].

The spectra of 2-MeBI **7** in CDCl_3_ showed a set of four sharp signals for the carbons of the BI moiety, characteristic of a tautomeric equilibrium. In contrast, the spectrum of 2-MeBI **7** in DMSOd_6_ showed a set of seven signals, characteristic of a non-symmetric BI ring. Because of the very slow dynamic phototropic equilibrium, δC4 and δC7 (117.8/110.5) appeared as broad separated signals, which indicate that **7a** is anchoring to differentiate both pyridine and pyrrole nitrogen atoms. In these conditions, C3a and C7a are shifted 5.0 and 4.0 ppm to higher frequencies, respectively, compared with the spectra obtained in CDCl_3_.

In the case of 2-acylBIs **8**, the formation of an NH∙∙∙O hydrogen bond (HB) anchors the molecule and the spectra showed a set of seven signals, being δC4 and δC7 at 122.0 and 112.0 ppm, respectively, characteristic of pyridine and pyrrole nitrogen atoms.

The 2-carboxylateBI **9** showed four signals of symmetrical BI in the ^13^C NMR spectra, whose tautomeric equilibrium **9a** = **9b** = **9c** = **9d** is demonstrated with the presence of the averaged value of C4/C7 signal in 116.3 ppm (about 60% pyridine nitrogen atom character **9a** = **9b**). This result is in accordance with the X-ray diffraction structure of this molecule found as a zwitterionic form in the solid state [36,37].

**Table 5 molecules-27-06268-t005:**
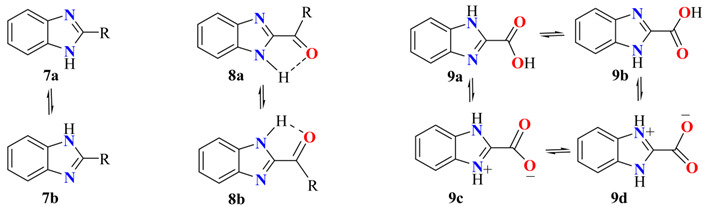
Effects of 2-subtituted groups on the benzimidazole ring.

Compound	C2	C3a	C4	C5	C6	C7	C7a	Solvent	Ref.
**7** (R = Me)	151.3 151.2	138.3 143.5	114.4 117.8	122.2 120.7	122.2 121.2	114.4 110.5	138.3 134.3	CDCl_3_ DMSOd_6_	[38] [33]
**7** (R = Et)	157.4	139.4	114.7	122.7	122.7	114.7	139.4	DMSOd_6_	[33]
**7** (R = *^n^*Pr)	155.6	138.7	114.6	122.0	122.0	114.6	138.7	CDCl_3_	[33,39]
**7** (R = *^i^*Pr)	142.3	135.4	118.9	120.9	121.7	110.8	134.4	DMSOd_6_	[33]
**7** (R = Bn)	151.8	133.2	114.0	125.8	125.8	114.0	133.2	CDCl_3_	[40]
**7** (R = Ph)	151.8	144.4	119.4	123.0	122.2	111.8	135.5	DMSOd_6_	[31]
**8** (R = Me)	147.7	143.5	122.1	123.8	122.1	112.1	126.6	CDCl_3_	[41]
**8** (R = Ph)	147.7	143.9	122.3	126.5	123.8	112.0	133.9	CDCl_3_	[39]
**9**	144.9	136.6	116.3	124.8	124.8	116.3	136.6	DMSOd_6_	

## 5. Iminothiol-Thioamide Tautomerism and Mesomers in 1,3-Benzoheterazolidine-2-thiones

### 5.1. Tautomers and Mesomers in Benzothiazolidine Tione

Benzazolidine-2-thiones (BZTs) **10**–**13** are constituted by a benzene ring fused to oxazolidine-, thiazolidine-, or imidazolidine-2-thione rings. The tautomeric equilibria between thioamide (NHC=S) and iminethiol (NCSH), thione = thiol, in compounds **10**–**13** are shown in the scheme on Table 6. In these tautomeric equilibria, a hydrogen atom is exchanging from the endocyclic nitrogen to the exocyclic sulfur atom, with the consequent transformation of the endocyclic nitrogen atom from the pyrrole to pyridine character. This phenomenon has been described in the solid state [8,42,43,44,45,46]. Thermodynamic studies in solution have concluded that tautomeric equilibrium is shifted to the thione form in substituted benzene compounds [47,48,49]. The BZTs **10**–**13** and their corresponding 2-S-Methyl-benzazoles **14**–**17** (TMBZ) are interesting compounds with biological activity [50].

It has been established that compound **12** predominantly exists as the thione (BIT) form in polar solvents and in the solid state [8]. The proton is bonded to the more electronegative atom (NH) in the molecule. The formation of a strong NH hydrogen bond with the solvent or another molecule of **12** is preferred instead of the S-H bond [11]. Model compounds, whose tautomeric equilibria has been locked with an S-Me, compound **14-17**, have been used to qualitatively estimate the tautomeric ratio. In addition, the formation of metallic complexes to stabilize the tautomers has also been reported [55,56].

In 1986, Balestero et al. quantitatively studied the thioamide/iminothiol tautomeric equilibria in BZTs **10**–**13** using the ^15^N-NMR spectroscopy. The advantage of this nucleus is its nuclear spin of ½, narrow lines, and a large chemical shift nitrogen difference between the tautomers (about 100 ppm) [11]. In DMSO solution, BZTs **10**–**13** were mainly in the thioamide form (NHC=S): BOT **10** (100%), BTT **11** (99%), BIT **12** (92%), and 1-MeBIT **13** (98%). However, because this tautomeric ratio depends on the temperature, concentration, and solvent polarity, these values could change with the experimental conditions.

The analysis of thioamide/iminothiol tautomerism has been performed on the bases of the reported ^13^C-NMR spectra of BZTs **10**–**13** and their respective TMBZs **14**–**17**; Table 6. In the ^13^C-NMR spectra of BIT **12**, four signals were found in DMSOd_6_, in agreement with the thione tautomer **12a**. The X-ray diffraction structure reported for **12a** agrees with these results [42].

On these bases, the observed C4/C7 chemical shift at 109.6 ppm of BIT **12a** is a good reference value for the pyrrole tautomer δC7_ref_. On the other hand, δC4 of TMBI **16** at −65 °C can be used as the pyridine reference. Despite that a set of seven signals appeared in the spectra at −65 °C, Table 7, the system is not completely frozen because δC4_obs_ and δC7_obs_ were at 117.8 and 110.9 ppm, respectively [57]. However, these data can be extrapolated to determine δC4_ref_; for this, we used the δC7_ref_ reference value (109.6 ppm) to calculate the difference: δC7_obs_ − δC7_ref_ (110.9 − 109.6 = 1.3 ppm), which was used to determine the δC4_ref_ = δC4_obs_ + 1.3 = 117.8 + 1.3 = 119.1 ppm.

The thioamide/iminothiol tautomerism of BZTs **10**–**13** can be estimated with the equation proposed by Balestrero et al. [11] for ^15^N-NMR data, applied to ^13^C chemical shift using values of δC4_ref_ and δC7_ref_, obtained from reference model compounds weighted by the respective molar fractions, as stated through previously described Equations (1)–(3). 

The spectra of TMBI **16** were obtained in CDCl_3_ and DMSOd_6_ at 27 °C. In both cases, only four signals were in the spectra, indicating the presence of a symmetric structure in tautomeric equilibrium; Table 7. The use of Equation (2) to calculate each tautomeric ratio indicates that compound **16** is present in 57% in CDCl_3_ and 58% in DMSOd_6_ as the NH tautomer. The spectra of compound **16** obtained in DMF-d_7_ at −65 °C showed a set of six separated signals, and C4 and C7 were observed in 117.8 and 110.9 ppm, respectively. In this case, from δC4_obs_ = 117.8 ppm, the N tautomer is calculated in 86.3%, whereas from δC7_obs_ = 110.9 ppm, the NH tautomer is calculated in only 13.7%. The same references were used to calculate the tautomeric ratio of compounds **10**–**17**; Table 6.

### 5.2. C4/C7 ^13^C NMR Chemical Shifts of 1,3-Benzoheterazolidine-2-tiones in Iminothiol-Thioamide Tautomerism and Benzimidazolidine-2-selone Mesomerism

Both X and N heteroatoms of BZTs **10**–**13** exert the respective shielding electronic effect—X atom on C7 and C5, and N atom on C4 and C6, as represented in Figure 4.

In the ^13^C-NMR spectrum of BOT **10** in CDCl_3_, the oxygen and nitrogen atoms exert a shielding electronic effect, shifting C7 to 110.5 ppm and C4 to 110.2 ppm. In this case, δC4_ref_ = 120.5 referenced to BO and δC7_ref_ = 109.6 ppm referenced to BIT **12**, and using Equation (2) for compound **10** with δC4_obs_ = 110.2 resulted in 95% of the pyrrolic character. Then, the endocyclic nitrogen atom is mainly tautomer **10a**, Table 6, near to that calculated by *Balestrero* et al. in 1986 [11]. In DMSOd_6_, δC4 appeared at higher frequencies than in chloroform, because of the change in the tautomeric mixture of the pyrrole **10a** (67%) and pyridine **10b** (33%) character, evaluated through δC4_obs_ = 113.2 ppm.

In the same solvent, for BTT **11**, no considerable shielding electronic effect of sulfur atom was observed on C7 (121.6 ppm) compared with that of BT **2** (123.6 ppm). However, the nitrogen atom shifted C4 to 112.3 ppm compared with 121.8 ppm in BT **2** (δC4Nr), allowing to estimate 77% of pyrrole tautomer **11a**. In the case of the TMBZs **14**, **15**, and **17**, their C4 observed values were found in 118.1, 121.7, and 117.4 ppm, respectively. Taking into account the C4 (119.1 ppm) and C7 (109.6 ppm) reference values, the proportion of their mesomeric form **b** with pyrrole negative nitrogen atoms was calculated as a minor proportion in 10.5%, 2.4%, and 17.9%, respectively, in DMSO-d_6_; Table 6.

Braun et al. [59] reported the synthesis of 1-EtBIT, analogous to compound **13**. A set of seven signals were found in the spectra of this compound in DMSOd_6_; the chemical shifts of C3a and C7a are at 131.9 and 130.7, C4 and C7 are at 109.6/109.3, C5 and C6 in 122.2 and 122.1, and C2 = **S** at 167.6 ppm, respectively. We can see that the two nitrogen atoms are in pyrrole form, indicating the NH tautomer to be present in 100%.

Palmer et al. [58] synthetized 1-alkylbenzimidazolidine-2-selenones (1-AkBISe) **18** and **19** and their corresponding diselenides **20** and **21**. In the X-ray diffraction structural analysis of 1-MeBISe **18** and 1-*^t^*BuBISe **19**, the hydrogen atom was located on the N atom (NH), demonstrating the structure to be the selenone tautomer. The density functional theory (B3LYP) calculations indicated that selenones are more stable than the selenol tautomers. Moreover, these data indicated that the selone is best represented as a C^+^–Se^-^ zwitterion instead of as a C=Se double bond. This was confirmed in the ^13^C-NMR spectrum obtained in DMSOd_6_ (Table 7); both N1 and N3 atoms in 1-MeBISe **18** are mainly as pyrrole, producing the protecting effect on C4/C7 (110.0 ppm). Mesomer **18a** was calculated in 95.8% in equilibrium with the mesomer **18b** (4.2%). On the other hand, the *^t^*BuBISe **19** showed separate signals for C4/C7 (114.4/109.9 ppm), indicating, on this basis, the reference values (119.1 and 109.6 ppm) and, for C4_obs_, the presence of mesomeric forms **19a**/**19b** in a 49.5/50.5 ratio. Eventually, mesomers **18c** and **19c** are present in a minor proportion. In the case of diselenides **20** and **21**, C7 in 120.4 and 118.8 ppm is indicative that NR atoms are principally the pyridine type. Then, their δC4 chemical shifts at 110.5 and 114.1 ppm, respectively, show the pyridine character of N3. On the basis of these observed chemical shifts, mesomers **20a**/**20b** was calculated in a 90.5/9.5% ratio and **21a**/**21b** in a 52.7/47.3% ratio. The presence of negative charge on N3 in tautomers **20b** and **21b** was confirmed because C3a is shifted approximately 13.0 ppm to higher frequencies, as the same carbon atom (~147.0 ppm) was shifted 8.0 ppm in the deprotonated BI spectrum in Table 3.

### 5.3. Mesomerism in N1-acyl-1,3-benzoheterazolidine-2-thiones

Some examples about N1-acyl-BZT **22**–**25** were found in the literature and the ^13^C NMR data are listed in Table 8. In the case of compound **22**, δC4 at 115.8 ppm indicates the presence of the mesomers **22a**/**22b** in approximately a 65/35 proportion. However, compound **23** shows the presence of two separated signals for C4 and C7 in 114.3 and 108.5 ppm, respectively. The last signal corresponds to a pyrrole N1H atom (tautomer **23a**). The signal in 114.3 ppm indicates that this molecule is in the mesomeric forms **23a** and **23b**, which contribute to the whole structure in approximately a 50/50 proportion. The same mesomers in approximately a 67/43 proportion were observed for compound **24a**/**24b** (C4 in 115.0 ppm, C7 in 109.4 ppm). In compound **25**, four signals of a symmetric BI ring were observed on the spectra C4/C7 in 113.2 ppm, indicating the mesomer **22a** is present in a major proportion.

### 5.4. Tautomeric Equilibrium in Omeprazole

In 2004, Claramunt et al. [4] reported the ^1^H and ^13^C NMR spectrum of Omeprazole in THF-d_8_ at −78 °C. The integration of ^1^H NMR signals led to the 37/63 ratio of the two tautomers **26a**/**26b**; Table 9. However, on the basis of the reported ^13^C NMR data, Table 9, we proposed the presence of two tautomeric pairs in equilibrium **26a**/**26b** and **26c**/**26d**, which form six-membered NH∙∙∙N and five-membered NH∙∙∙O hydrogen bonding interactions, respectively. Each tautomeric pair is represented by one of the two sets of signals reported. The δC7_obs_ value at 113.1 ppm and Equation (2) allowed us to estimate that the pair **26a**/**26b** represents 63% of the four tautomers, with **26b** being in a major proportion because C7 (101.1 ppm) is affected by both OMe and N_pr_ atoms. The remaining 37% corresponds to the pair **26c** = **26d**, where δC7 in 94 ppm represents the presence of **26c** as the main tautomer. This was considered because, in this case, C7 in **26c** suffers the electronic effect of both MeO oxygen atom and NH pyrrole nitrogen. On the other hand, the δC4 at 121.6 ppm is characteristic of a N_pd_ electronic effect.

## 6. Conclusions

In BI derivatives, the C4/C7 chemical shift values are sensitive to the phototropic tautomeric equilibrium: N3=C2-N1-H/H-N1=C2-N3. In general, these carbon atoms are shifted from 120 to 110 ppm, which involves the change in the nitrogen atom character from pyridine N3 to pyrrole N1-H, increasing the shielding electronic effect on C3a and C7 in the whole process.

In the protonated BI (HBI^+^), no electronic effect is induced by the =N-H^+^ nitrogen atom on C4/7; however, an inductive electronic effect shifts C3a/7a to low frequencies. The N-pyridine nitrogen atom, in deprotonated benzimidazole (BI^−^), exerts a protective electronic effect on C4/7 and C5/6, and deshields C2 and C3a/7a, shifting them to high frequencies.

In N-AkBIs, both =N3 pyridine nitrogen (δC4 = 120 ppm) and N-R pyrrole nitrogen atom (δC7 = 110 ppm) are present in the molecule. 

On the other hand, any carbonyl group bonded to the pyrrolic nitrogen atom diminishes the electronic delocalization into the aromatic system, decreasing the electronic effect on C7. The N-phenyl ring was found to be perpendicular to the BI plane when the spectra is obtained in CDCl_3_. Meanwhile, in DMSOd_6_, the phenyl ring is in the same BI plane with the consequent deshielding effect on C7.

Upon increasing the alkyl chain in 2-AkBIs in CDCl_3_, C3a/7a and C4/7 appeared as broad signals with tendency to disappear (coalescence). An acyl group bonded to C2 in BI favors the NH tautomer owing to an N-H∙∙∙O hydrogen bonding formation.

The C4 chemical shift of BIT (δC4_ref_ = 119.1 ppm) and the corrected C4 chemical shift of 1-MeTMBI (δC4_ref_ = 109.6 ppm) were used as references in Equation (2) to calculate the tautomeric ratio in BI derivatives.

The concepts for BIs can be applied to the nitrogen atom in benzoxazole and benzothiazole derivatives; for instance, in the case of BZTs, C4 and C7a were found to be shifted in about 10.0 and 4.0 ppm, respectively, to low frequencies when the nitrogen atom change from the pyridine to pyrrole type. On the other hand, alkyl groups bonded to a BOTs or BTTs nitrogen atom, C4, suffer the whole protective electronic effect of N1-R pyrrole nitrogen atom, shifting C4 to low frequencies (110 ppm).

The proposal criterion about the electronic effects on the ^13^C chemical shifts of the heteroatoms on the aromatic ring in benzazoles can be used to the correct assignment of any substituted benzazole derivatives as well as to estimate the proportion of the tautomeric species.

## Data Availability

Not applicable.

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
