# Peer review of "13C-NMR Chemical Shifts in 1,3-Benzazoles as a Tautomeric Ratio Criterion"

_molecules, 2022, doi:10.3390/molecules27196268_

Round 1

Reviewer 1 Report

This paper takes literature 13C NMR solution data on a range of molecules that contain benzimidazole fragments to develop the approach for analysing such data to determine the 1,3-tautomeric equilibrium. The way the paper is presented shows a good knowledge of the literature and for the most part presents a good reanalysis of that data to add some value which makes the paper worth publishing eventually. The literature is well referred over a range of the relevant compounds and gives reference 1H and 15N work as well some solid-state NMR work, although this is all context for the 13C work here. The key observation is the shift difference between the C4 and C7 positions depends on which tautomer is present. Then depending on the exchange rate the data can be analysed using different approaches to determine the tautomeric ratio. The paper describes well the different approaches to examining this problem which makes it a good initial introduction to the non-expert reader. In this spirit the authors then need to think about how they explain some of their analysis. For example Eqs 1 and 2 are introduced on p8 to explain the analysis of the data in Table 7 which is fine. However on p2 there is analysis of the C4/C7 data Table 3 to get the tautomeric ratio. There needs to be a detailed explanation here, with inclusion of the relevant equations to explain the ratios that have been derived. For example it is not clear that if a shift position of 116.3 ppm corresponds to 63% pyridine character, then 118.3 ppm corresponds to 91.6%?

Hence the paper needs to be looked at to ensure that all readers are able to follow the approach used by the authors. I have indicated above that I think the paper is publishable eventually. However in addition to ensuring that some of the arguments are more clearly developed there are several minor points that mean the paper in its current form is not publishable. There are far too many typographical errors throughout which spoil the paper and a spell checker would eliminate many of them along with a careful re-reading. There is then a mixed format to the insertion of references – most are included in [   ], but several (e.g. p2, line 62 and line 73) are included as a superscript. For some tables it is immediately apparent which publications the data has come from and for others it is not. On p11 the final table (Table 9) there is no table caption.

Author Response

Referee 1

This paper takes literature 13C NMR solution data on a range of molecules that contain benzimidazole fragments to develop the approach for analysing such data to determine the 1,3-tautomeric equilibrium. The way the paper is presented shows a good knowledge of the literature and for the most part presents a good reanalysis of that data to add some value which makes the paper worth publishing eventually. The literature is well referred over a range of the relevant compounds and gives reference 1H and 15N work as well some solid-state NMR work, although this is all context for the 13C work here. The key observation is the shift difference between the C4 and C7 positions depends on which tautomer is present. Then depending on the exchange rate the data can be analysed using different approaches to determine the tautomeric ratio. The paper describes well the different approaches to examining this problem which makes it a good initial introduction to the non-expert reader. In this spirit the authors then need to think about how they explain some of their analysis. For example Eqs 1 and 2 are introduced on p8 to explain the analysis of the data in Table 7 which is fine. However on p2 there is analysis of the C4/C7 data Table 3 to get the tautomeric ratio. There needs to be a detailed explanation here, with inclusion of the relevant equations to explain the ratios that have been derived. For example it is not clear that if a shift position of 116.3 ppm corresponds to 63% pyridine character, then 118.3 ppm corresponds to 91.6%?

R: Thank you for this valuable suggestion. The equations 1-3 were re-written according to Figure 2 and simplified, then were positioned before Table 3. A detailed explanation was also included as requested.

Hence the paper needs to be looked at to ensure that all readers are able to follow the approach used by the authors. I have indicated above that I think the paper is publishable eventually. However, in addition to ensuring that some of the arguments are more clearly developed there are several minor points that mean the paper in its current form is not publishable. There are far too many typographical errors throughout which spoil the paper and a spell checker would eliminate many of them along with a careful re-reading. There is then a mixed format to the insertion of references – most are included in [   ], but several (e.g. p2, line 62 and line 73) are included as a superscript. For some tables it is immediately apparent which publications the data has come from and for others it is not. On p11 the final table (Table 9) there is no table caption.

R: Thank you for your comments. (1) The text was revised, and typographical errors corrected. (2) References were included in brackets. (3) We included the bibliographic reference in all Tables to make them clear. (4) The caption on Table 9 was included.

Reviewer 2 Report

This paper reviews 13-C NMR for benzimidazole (BI) derivatives with focus on N1/N3H tautomeric equilibrium. According to the abstract, lines 21 and 24, the study appears to go beyond a review, with new conclusions and analyzes being reported. This should be clarified for readers, whether the present study is a literature review or a regular paper, with new results and conclusions. If new results and discussion are to be presented, a methodology section would be important, describing in detail how the data were interpreted and what differs from the analysis available in the literature. It is not common for reviews to include sections like 'Results' or 'Analysis and Results', which I suggest removing, keeping only the subsections. However, if the paper was submitted as a regular paper, the 'Results and Discussion' section must be included in addition to the 'Methodology'. In fact, for this Reviewer, the way the text was organized is not easy to follow. This first comment is not due to merit, but to the organizational structure of the text. Next, I will make some specific comments and suggestions.

1) If the N1/N3H equilibrium is the focus, why HBI+ and BI- derivatives were analyzed? For these derivatives, N1/N3H tautomerism does not occur.

2) Abstract, line 17. ‘…The intensity of the 1H-NMR signals has been used to determine the tautomeric ratio….’. I could not see where the signal intensity was discussed, only the chemical shifts are reported and discussed.

3) Page 2, line 87. ‘…in this work, we proposed a method to calculate in solution the tautomeric ratio in BZs on the basis of the electronic effect of pyrrole nitrogen atom on C4/C7 chemical shift resonances…’. This statement is typical of a regular paper where new data and analysis are being reported for the first time.

4) Sec. 2.1, page 3, line 95. What kind of molecule is this? Which carbon is Ci and where the substituent is included? Why this series of molecules is important for the tautomerism issue?

5) Page 3, line 102. “…On the other hand, the chemical shift of C4 in the dibenzofused heteroaromatic compounds 1-3 (Table 1), appears at 111.6, 110.8 and 122.9 ppm for X = O, NH and S, respectively…” The order of values is different from that given in Tab. 1,  which X=NH shows C4 chemical shift at 122.9 ppm. I think it is inverted in Tab. 1.

6) Page 4, line 134. I understand that if C4 and C7 show different chemical shift a single tautomer is dominant and the equilibrium should be slow. In this part of text, the chemical shifts for C4 and C7 are different in solution and solid state (Tab. 2), therefore, the equilibrium should be slow at the experimental conditions. What is different in solution and solid state?

7) Page 4, line 138. ‘…In BIs, the tautomeric species are the result of intermolecular proton transfer from N1 to N3…’. It should be ‘intramolecular’ proton transfer. Besides, what do you mean by ‘…observed C4/C7 chemical shifts can be used as a criterion for tautomeric ratio determination….’? Do you mean average value of chemical shifts or ratio of chemical shifts?

8) Page 4, line 153. The calculation of %Npr should be explained in Fig. 2. The relationships %Npr(120-dobs)=%Npd(dobs-110) and %Npr+%Npd=1 could be included in the Fig. 2 caption. Still regarding this point, how the value 116.3 ppm was obtained? Does it mean the equilibrium is fast and only one signal is observed?

9) Page 4, line 157. The data in Tab. 3, the tautomerism exists only for 3a/3b. For the other species, both N1 and N3 are protonated or deprotonated. For 3a/3b equilibrium, why the ratio is not 50%, once both forms are equivalent?

10) There are two different ‘Scheme 1’.

Author Response

Referee 2

This paper reviews 13-C NMR for benzimidazole (BI) derivatives with focus on N1/N3H tautomeric equilibrium. According to the abstract, lines 21 and 24, the study appears to go beyond a review, with new conclusions and analyzes being reported. This should be clarified for readers, whether the present study is a literature review or a regular paper, with new results and conclusions. If new results and discussion are to be presented, a methodology section would be important, describing in detail how the data were interpreted and what differs from the analysis available in the literature. It is not common for reviews to include sections like 'Results' or 'Analysis and Results', which I suggest removing, keeping only the subsections. However, if the paper was submitted as a regular paper, the 'Results and Discussion' section must be included in addition to the 'Methodology'. In fact, for this Reviewer, the way the text was organized is not easy to follow. This first comment is not due to merit, but to the organizational structure of the text. Next, I will make some specific comments and suggestions.

R: Thank you for your comments and suggestions. (1) We clarified to the readers that this paper is a review of literature 13C-NMR solution data, which we used to develop an approach to estimate the 1,3-tautomeric equilibrium. This precision is included in the abstract and introduction sections. (2) The paper was re-structured with the aim of improve its reading.

1) If the N1/N3H equilibrium is the focus, why HBI+ and BI- derivatives were analyzed? For these derivatives, N1/N3H tautomerism does not occur.

R: Thank you for your observation. This section was re-organized and the discussion of the mentioned species deleted.

2) Abstract, line 17. ‘…The intensity of the 1H-NMR signals has been used to determine the tautomeric ratio….’. I could not see where the signal intensity was discussed, only the chemical shifts are reported and discussed.

  1. This sentence was misplaced and deleted from the abstract.

3) Page 2, line 87. ‘…in this work, we proposed a method to calculate in solution the tautomeric ratio in BZs on the basis of the electronic effect of pyrrole nitrogen atom on C4/C7 chemical shift resonances…’. This statement is typical of a regular paper where new data and analysis are being reported for the first time.

R: This sentence was deleted and the following was included: “13C-NMR solution data was retrieved from literature and used to develop an approach to estimate the 1,3-tautomeric equilibrium”

4) Sec. 2.1, page 3, line 95. What kind of molecule is this? Which carbon is Ci and where the substituent is included? Why this series of molecules is important for the tautomerism issue?

R: Thank you for this observation. We consider that a brief revision of the substituents effects on the benzene chemical shifts is necessary as introduction, then the following statemen was included:

“For clarity reasons, a brief introduction about the inductive and electronic effects of heteroatoms (N, O, S) on the 13C chemical shifts (d) of the benzene ring [26], is required.” We also indicate the meaning of Ci, Co, Cm and Cp as the ipso, ortho, meta and para positions, respectively.

5) Page 3, line 102. “…On the other hand, the chemical shift of C4 in the dibenzofused heteroaromatic compounds 1-3 (Table 1), appears at 111.6, 110.8 and 122.9 ppm for X = O, NH and S, respectively…” The order of values is different from that given in Tab. 1, which X=NH shows C4 chemical shift at 122.9 ppm. I think it is inverted in Tab. 1.

R: Thank you for this observation. This mistake was corrected:

“In this sense, the dC4 in the dibenzofused heteroaromatic compounds 1-3 (Table 1), appears at 111.6, 110.8 and 122.9 ppm for X = O, NH and S, respectively [26]. As expected, the inductive effect of the heteroatom on Ci shifts d to high frequencies, in agreement with the electronegativity of the heteroatom.”

6) Page 4, line 134. I understand that if C4 and C7 show different chemical shift a single tautomer is dominant and the equilibrium should be slow. In this part of text, the chemical shifts for C4 and C7 are different in solution and solid state (Tab. 2), therefore, the equilibrium should be slow at the experimental conditions. What is different in solution and solid state?

R: In solution the observed chemical shifts are isotropic and the values are averaged. Instead, in the solid there are effects of the asymmetric unit and polymorphism, among others. In fact, at least five polymorphs of BI are reported, and whose 13C-CPMAS are expected to be different, please see the following reference.

  1. Cabildo, R. M. Claramunt, F. J. Zuñiga, I. Alkorta, J. Elguero, Crystal and molecular structures of two 1H-2-substituted benzimidazoles, Z. Kristallogr. 2015; 230(6): 427–438.

7) Page 4, line 138. ‘…In BIs, the tautomeric species are the result of intermolecular proton transfer from N1 to N3…’. It should be ‘intramolecular’ proton transfer. Besides, what do you mean by ‘…observed C4/C7 chemical shifts can be used as a criterion for tautomeric ratio determination….’? Do you mean average value of chemical shifts or ratio of chemical shifts?

R: In most of the literature, the prototropic equilibrium in BIs is considered as intermolecular as we do. On the other hand, the average value of the C4 and C7 chemical shifts is the value herein used as a criterion for the estimation of tautomeric equilibrium. 

8) Page 4, line 153. The calculation of %Npr should be explained in Fig. 2. The relationships %Npr(120-dobs)=%Npd(dobs-110) and %Npr+%Npd=1 could be included in the Fig. 2 caption. Still regarding this point, how the value 116.3 ppm was obtained? Does it mean the equilibrium is fast and only one signal is observed?

R: This section was re-written as follows:

“The chemical shift reference values for the estimation of the tautomeric equilibrium were set using the 1-methyl-benzimidazole (MeBI) as a model compound. In the case of MeBI (4), the tautomeric equilibrium is inexistent therefore a set of seven narrow signals are observed in CDCl3. The dC4 appears at 120.4 ppm, characteristic of a pyridine like ni-trogen (Npd) effect, whereas the dC7 at 109.5 ppm is characteristic of a pyrrole like nitrogen (Npr) effect. In DMSOd6, dC4 and dC7 values are at 119.2 and 110.1, respectively. Then a good approximation for 100% of pyridine like character of dC4 and 100% of pyrrole like character of dC7 are 120.0 and 110.0 ppm, respectively. Furthermore, the tautomeric pro-portion in BIs can be calculated on a base equal to 120.0 ppm for dC4ref (Npd), and 110.0 ppm for dC7ref (Npr) as reference values, Figure 2.”

9) Page 4, line 157. The data in Tab. 3, the tautomerism exists only for 3a/3b. For the other species, both N1 and N3 are protonated or deprotonated. For 3a/3b equilibrium, why the ratio is not 50%, once both forms are equivalent?

R. For BI, the dobs value is 116.3, then the estimation is 63% of pyridine character.

It can be because of several molecules of benzimidazole are forming chains and they have two diferents nitrogens (hydrogen donor and acceptor) the acceptor nitrogen  is more weakly  bonded ( more pyridinic nitrogen carácter)  that the donor nitrogen .

In the case of dobs for protonated BI (118.3 ppm), a little effect of positive charge on nitrogen atoms shifts this signal to high frequencies in approximately 2.0 ppm. On the contrary, any effect of the negative charge on nitrogen (pyrrolic nitrogen) in deprotonated BI was observed.

10) There are two different ‘Scheme 1’.

R: The numbering of Tables, Figures and Schemes was checked and corrected.

Reviewer 3 Report

page 1, line 17 spelling check

page 1, line 26 researchers spelling
page 1, line 31  page , line 57 overall spelling needs to be checked.   a very interesting study, as in the protein binding cavity the phenyl rings undergo a pi-stacking interaction, how do that change the ppm of these BZs compounds.    The selenium compounds will be an interesting topic for th readers.   like the diselenides, is there a study to add for disulfides, as these are more relevant for protein chemists.    is it possible to present in a figure of all 13C signals for the compounds to have a general idea on utset.

Author Response

Referee 3

page 1, line 17 spelling check

page 1, line 26 researchers spelling
page 1, line 31  page , line 57 overall spelling needs to be checked.   a very interesting study, as in the protein binding cavity the phenyl rings undergo a pi-stacking interaction, how do that change the ppm of these BZs compounds.    The selenium compounds will be an interesting topic for the readers.   like the diselenides, is there a study to add for disulfides, as these are more relevant for protein chemists.    is it possible to present in a figure of all 13C signals for the compounds to have a general idea on utset.

R. Thank you for your comments and observations. The document was checked for spelling mistakes and mistyping which were corrected.

-a very interesting study, as in the protein binding cavity the phenyl rings undergo a pi-stacking interaction, how do that change the ppm of these BZs compounds.   

-The selenium compounds will be an interesting topic for the readers.   like the diselenides, is there a study to add for disulfides, as these are more relevant for protein chemists. 

R. We did not found BI diselenides

-is it possible to present in a figure of all 13C signals for the compounds to have a general idea on utset

R. We think tables are best.

Round 2

Reviewer 1 Report

My original report raised issue mainly around presentation and more clarification. These have largely been done. Given thsi and the other reviews particularly in response to Reviewer 2 I am content to recommend publication.

Reviewer 2 Report

My previous comments have been carefully considered by the authors with clear point-to-point answers. The manuscript was improved, which I am pleased to recommend for publication in this revised form.